# Patellar Tendon Elasticity and Temperature Following after a 448 Kilohertz Radiofrequency Intervention on Active Healthy Subjects: An Open Controlled Clinical Trial

**DOI:** 10.3390/diagnostics13182976

**Published:** 2023-09-18

**Authors:** Maria Cuevas-Cervera, Daniel Aguilar-Nuñez, María Aguilar-García, María Carmen García-Ríos, Ana González-Muñoz, Santiago Navarro-Ledesma

**Affiliations:** 1Department of Physiotherapy, Faculty of Health Sciences, Campus of Melilla, University of Granada, Querol Street, 5, 52004 Melilla, Spain; maaricuevass@correo.ugr.es (M.C.-C.); maguilar.fisioterapia@gmail.com (M.A.-G.); snl@ugr.es (S.N.-L.); 2Biomedicine PhD Program, Faculty of Health Sciences, University of Granada, Av. de la Ilustración, 60, 18071 Granada, Spain; 3Department of Nursing and Podiatry, Faculty of Health Sciences, University of Malaga, 29071 Malaga, Spain; daguilarn.tic@gmail.com; 4Department of Physiotherapy, Faculty of Health Sciences, University of Granada, Av. de la Ilustración, 60, 18071 Granada, Spain; mcgrios@ugr.es; 5Clinica Ana Gonzalez, Avenida Hernan Nuñez de Toledo 6, 29018 Malaga, Spain

**Keywords:** patellar tendon, radiofrequency, elastography, termography

## Abstract

The purpose of this study was to analyze the changes in the elasticity and temperature of the patellar tendon produced by the application of a radiofrequency at 448 kHz (CRMR) just after and 7 days after the intervention. An open controlled clinical trial was used with participants being recruited from a private clinic. The experimental group (*n* = 22) received a 448 kHz CRMR treatment while the control group (*n* = 22) did not receive any type of intervention. Quantitative ultrasound strain elastography (SEL) and thermography were used to collect data from 4 different areas of the patellar tendon. These areas were measured at the start (T0), just after (T1), and seven days after (T2) the intervention. There were thermal changes immediately after the intervention (*p* < 0.001). In addition, when the measurements were collected just after the intervention and seven days after they were analyzed, significant changes (*p* < 0.001) in temperature were observed in the tendons of both groups. Finally, a low but significant association (r = 0.434, *p* < 0.04) was observed between the elastic properties of the tendon at its insertion in the patella and thermal changes just after the 448 kHz intervention.

## 1. Introduction

The patellar tendon is a linear structure composed of a superficial and a deep layer that runs in parallel from the patella to the tibia without direct muscle insertions [1]. It is capable of supporting large volumes of weight and is one of the tendons responsible for providing stability and extension of the knee [2,3]. Due to its morphology and adaptive properties, the tendon exhibits the capacity to respond to both intrinsic and extrinsic stimuli. The patellar tendon can undergo thickening and remodeling due to the pull that occurs by repeated forces [2].

Patellar tendinopathy is typically characterized by anterior knee pain, localized at the lower pole of the patella [4]. This tendinopathy is usually more common in athletes who play jumping sports such as basketball or volleyball [4], with 45% of elite athletes experiencing this type of pain [1]. The most common cause of patellar tendinopathy is repetitive load on the tendon, although there are theories that say there may be vascular, mechanical, or causes related to impingement [4]. Patellar tendon rupture, which is 0.6% of musculoskeletal injuries [5], is often accompanied by rupture of other structures such as the anterior cruciate ligament or collateral ligament.

The patellar tendon is traditionally assessed through a physical examination and an X-ray by means of which an attempt is made to find differences between the different possible diagnoses. This type of testing may be limited so ultrasound is presently used as it can provide additional information [6]. Currently, elastographic techniques evaluate stiffness, and tissue tension, providing a higher-quality diagnosis [7]. Recent studies encourage the combination of traditional ultrasound diagnosis with elastographic techniques to increase diagnostic quality [7]. The presence of a soft patellar tendon has been associated with tendinopathy, pain, and functional impairment [8,9]. However, tendon stiffness has been seen to be greater in those athletes who, apart from presenting patellar tendinopathy, had calcifications than those who did not [10].

On the other hand, thermography is a non-invasive imaging tool, that does not need contact, is safe, and measures physiological variables and body temperature control [11]. The information obtained results from the hyperthermic and hypothermic responses suffered by the skin. This technology is able to measure the relationship between body temperature and muscle activation [12]. Thermography has proven to be a great tool in preventing injuries and disorders since due to its use related factors such as inflammatory processes or secondary traumas can be observed, thus reflecting the asymmetry among various body regions [13].

Radiofrequency at 448 kHz is considered to be a non-invasive technique capable of inducing hyperthermia in different tissues with the ability to have an effect on deep structures, such as muscles and joints [14]. The therapeutic purpose of heat is to relieve pain and inflammation, as well as to promote tissue healing by accelerating or slowing cellular activity, promoting vasodilation, and promoting local tissue circulation [14,15,16]. Different physiological mechanisms that repair pain and inflammation are released under the effect of heat [14]. Heat can change the nature of tissue connectivity. It can alter the properties of tendons, ligaments, and some muscles, increasing extensibility and reducing tone and spasms. The physiological effect will depend on the level of exposure to which the different tissues are subjected [14].

The application of capacitative-resistive monopolar radiofrequency (CRMR) at 448 kHz in healthy subjects has been shown to be more effective than conventional thermotherapy at an increasing temperature in superficial and deep tissues and also blood flow in both skin and deep tissues, as well as increasing hemoglobin saturation [14,17]. Radiofrequency adaptations to hyperthermia are based on vasodilation, which causes increased local blood perfusion and reduced muscle tension and spasm, increasing oxygen and nutrients as well as accelerating the healing process [18]. CRMR shows an ability to improve pain, function, and quality of life in patients with musculoskeletal disorders, especially spinal disorders and knee osteoarthritis [17,19].

Studies have shown that the application of CRMR produces short, medium, and long-term effects on the elastic properties of the supraspinatus tendon in professional badminton players, where the elastic changes were measured by strain elastography (SEL) [20]. However, there are no other studies in this line, nor is it known what the physiological response of a load tendon, such as the patellar tendon, would be after an intervention of 448 kHz CRMR.

The objective of this study is to analyze the elastographic and thermographic changes that occur in the patellar tendon (PT) after the application of an intervention with 448 kHz CRMR in healthy physically active people and to analyze whether an intervention with CRMR in the PT of the dominant lower limb produces a thermal response in the PT of the non-dominant lower limb. The secondary objective will be to analyze the level of association between the elasticity of the PT and its temperature before and after an intervention with CRMR.

## 2. Materials and Methods

### 2.1. Design

An open-controlled clinical trial was used.

### 2.2. Setting

Subjects were enrolled in a private care practice located in Malaga, Spain. Prior to their participation, all individuals provided informed written and verbal consent, and their baseline demographic and clinical particulars were collected. A comprehensive understanding of the trial was conveyed to the participants via formal meetings and informative trial documents. The study, adhering to the principles of the Declaration of Helsinki, was duly registered on ClinicalTrials.gov under the registration identifier NCT05498987. Furthermore, the study protocol was granted approval by a Medical Research Ethics Committee, and its execution adhered to the CONSORT guidelines as outlined in [21,22].

### 2.3. Participants

The participants (14 men and 8 women) were selected according to inclusion and exclusion criteria (See Figure 1). The subjects were classified as physically active based on their customary level of physical activity according to guidelines laid out by Damsted et al. [23].

### 2.4. Inclusion Criteria

People between 18 and 65 years old, physically active, and with no pain or injury to the patellar tendon or knee were included in the study.

### 2.5. Exclusion Criteria

Participants presenting: (i) any form of pain or inflammatory process, (ii) neurological or orthopedic disease that could impair balance, hearing, and vision, or (iii) cognitive problems that could impair the ability to answer questions.

#### 2.5.1. Allocation

The dominant lower limb was considered as the intervention group and the non-dominant lower limb as the control group.

#### 2.5.2. Sample Size Calculation

The determination of the sample size was conducted using the EPIDAT software, a publicly accessible tool designed to support data management for epidemiologists and healthcare practitioners. Building upon insights from prior investigations concerning shoulder tendons [19,20], and considering an envisaged mean difference of 1.1 mm in tendon thickness, along with a corresponding standard deviation of 1.0 mm, and a predefined significance level of α = 0.05, a robust statistical analysis involving t tests projected a requisite sample size of *n* = 19 for each group, thereby ensuring a statistical power of 90%.

### 2.6. Intervention Description

#### 2.6.1. Experimental Group

Participants received one 448 kHz CRMR intervention. Each participant was positioned in a supine posture with their knees flexed at an angle of 20 degrees [18]. Utilizing the INDIBA^®^ Activ 8 apparatus, characterized by a peak power of 200 W and 450 VA, continuous radiofrequency electromagnetic radiation (CRMR) at 448 kHz was administered, consistent with previous investigations [19]. Electrodes composed of metal, which served as conductors, were employed to transmit capacitive (CAP) and resistive (RES) waves, facilitated by a coupling medium. To ensure a controlled application, the patient’s thermal perception was anticipated to reach a level of 8 out of 10. The patient was instructed to communicate the onset of this thermal perception, enabling the therapist to appropriately administer both CAP and RES modes while adhering to the manufacturer’s guidelines regarding thermal dosages. For a single session focused on the knee area, a total intervention duration of 15 min was allocated. Specifically, the CAP mode was implemented for 5 min, followed by the RES mode for 10 min. The return electrode was strategically positioned on the posterior aspect of the treated knee.

#### 2.6.2. Control Group

The participants’ non-dominant lower limbs did not receive any type of treatment; only measurements of the parameters of interest were made.

### 2.7. Outcome Measures

#### 2.7.1. Strain Elastography

A physiotherapist with 12 years of experience in musculoskeletal ultrasound imaging and 4 years of experience in SEL used a Logiq S7 with a 15 MHz linear probe (GE Healthcare, Milwaukee, WI, USA) to carry out all measurements. The evaluator carried out a protocol study before the research started, presenting an excellent ICC (>0.9). The measurements were performed following recommendations from previous studies [24,25] and with the patient lying supine with the knee flexed at 30 degrees. The patellar tendon was located using the kneecap as the upper border and the tuberosity of the tibia as the lower border [24]. The selected image featured 5 green bars, with this indicating the highest level of quality recommended by the inbuilt software in the computer [19,24]. Four circular 5 mm regions, which ranged from the kneecap to the tibial tuberosity, were used to calculate the SEL value along the patellar tendon (see Figure 2). The values shown ranged from 0 to 6, from the softest to the hardest [19].

#### 2.7.2. Thermography

Thermography measurements were performed following instructions from previous studies [11]. The room used for the measurements was equipped with an air conditioner to maintain the temperature at 22 °C (±1.5 °C) and the humidity at 40–60%. An X FLIR T420bx camera, which was placed 3 m from the participants who were in their underwear and barefoot, was used. The areas to be measured thermographically were selected through computerized image analysis and analyzed via the Thermohuman software. When analyzing the regions of interest, the minimum, maximum and mean temperature values were extracted to calculate the thermal data [11] (see Figure 3).

### 2.8. Statistical Analysis

The analytical processes were executed using SPSS^®^ Statistics version 21.0, developed by IBM in Chicago, IL, USA. Data distribution normality was assessed through the application of the Shapiro–Wilk test. A three-way repeated measures ANOVA was employed to investigate the clinical characteristics of the two groups across three distinct assessment points: baseline (T0), subsequent to the intervention (T1), and one week post-intervention (T2). For statistical inference, a significance threshold of *p* < 0.05 was adopted. To account for multiple comparisons, Bonferroni adjustments were implemented. The examination of correlations entailed Pearson’s and Spearman correlation coefficients, where correlations were categorized as weak (0.3 to 0.5), moderate (0.5 to 0.7), or strong (>0.7) in accordance with the magnitude of the correlation coefficient. In both correlation analyses and throughout the study, a significance level of *p* < 0.05 was considered indicative of statistical significance.

## 3. Results

Table 1 shows demographic characteristics. There were no significant differences in gender, age, weight, height, and patellar tendon elasticity between the compared lower limbs.

### 3.1. Differences in Patellar Tendon Elasticity and Temperature at Baseline after the Intervention and One-Week Follow-up

Comparisons between groups are described in detail in Table 2. There were statistically significant differences in the patellar tendon temperature observed when analyzing changes from immediately after the intervention program (T1) and one week after its end (T2).

### 3.2. Within-Group Differences in Patellar Tendon Elasticity and Temperature

Table 3 describes the within-group differences in patellar tendon elasticity and temperature at baseline (T0), after the intervention (T1), and at one week (T2) follow-up, with 95% CI. Significant differences in temperature were observed when analyzing changes from immediately after the CRMR intervention (T1) until the next measurement one week later (T2).

### 3.3. Association between Patellar Tendon Strain Elastography and Thermography

Table 4 shows the association between patellar tendon strain elastography and thermography at baseline (T0), after the intervention (T1), and at one-week (T2) follow-up in both groups. A low and significant association (r = 0.434, *p* < 0.04) was observed between the elastic properties of the tendon at measurement point 1 (insertion in the patella) and thermal changes immediately following the 448 kHz CRMR intervention.

## 4. Discussion

The results of the present study showed that a 448 kHz CRMR intervention causes thermal changes immediately after the radiofrequency application in physically active healthy people. In addition, there were significant changes in the temperature of both patellar tendons when analyzing the changes from just after the intervention to the next measurement that took place a week later. Finally, a low but significant association (r = 0.434, *p* < 0.04) was observed between the elastic properties of the tendon at measurement point 1 (insertion in the patella) and thermal changes just after the 448 kHz CRMR intervention.

It is difficult to compare our results with other studies since our study is the first to analyze the elastic and thermal changes of the patellar tendon after a CRMR intervention. The results of the study reflect that there is an immediate vascular response that occurs in the knee region following the application of 448 kHz CRMR, but that it does not affect the viscoelastic properties of the patellar tendon. The application of hyperthermia leads to the heating of deep muscle tissues, subsequently improving hemoglobin saturation. This leads to heightened blood flow in the deep superficial layers, dilation of blood vessels, and an elevation in temperature. These processes trigger reactions such as heightened blood perfusion, consequently leading to a rise in temperature. This phenomenon could elucidate the viscoelastic modifications in tendons measured by SEL. Interestingly, the thermal changes observed in the knee region when analyzing the differences between measurement times T1 and T2 may suggest the presence of a vascular response not only in the intervened limb (dominant) but also in the contralateral lower limb in future assessments, but this must be further studied. Accordingly, some studies show how an intervention in one limb can produce a response in the contralateral limb [26]. In this vein, given the potential for bilateral adaptations following a unilateral intervention. Delving into the underlying mechanisms through which these bilateral adjustments could influence symmetry in biomechanics, the responsiveness of tissues, and systemic physiological reactions offers valuable perspectives on their potential effects on sports-related achievements and injury prevention. This contemplation underscores the significance of evaluating how unilateral interventions could trigger systemic alterations that influence an athlete’s overall performance and predisposition to injuries.

The association found between elastic changes and temperature in the patellar tendon after the intervention coincides with the point of greatest injury frequency in this tendon [27], which may present histological changes at that point. This could have clinical consequences and should be researched in future studies. In this regard, a CRMR intervention together with an exercise program could be used to optimize the viscoelastic characteristics of the tendon at the insertional level which would favor mechanotransduction and tissue modeling [19], as has been suggested by other authors for the supraspinatus tendon in healthy athletes [19,20]. In addition, the results of this study could be used for clinical purposes in patients with patellar tendinopathy, or for athletes in terms of performance and short-term recovery, since modulation of the cellular inflammatory response after repetitive loading, as it occurs when training or competing, may also be benefited in those intervened with CRMR, but this is only one hypothesis and more studies are needed to confirm our findings. The phenomenon of dynamic loading is a common characteristic in both the maintenance of healthy tendon tissue and its pathological conditions. Tendon cells possess the intrinsic capability to perceive mechanical loads, which then trigger intricate mechanotransduction pathways at the molecular level [28]. While optimal mechanical loading is necessary for mature tendons to uphold and optimize their extracellular matrix structure, aberrant loading sets in motion an inflammation-driven tissue repair process that can potentially lead to disruption in extracellular matrix equilibrium and eventual tendon deterioration. The specific loading patterns and the inflammatory mechanisms that underlie tendon healing and pathological changes remain complex and not fully elucidated. Nonetheless, gaining a precise understanding of these loading and inflammatory processes is of paramount importance in advancing therapeutic strategies for effectively managing tendon pathologies [29,30].

Recent research has brought to light the influence of mechanical forces on the development of hypertrophic scarring. The mechanotransduction signaling pathway presents a promising avenue for potential intervention to mitigate scarring and foster tissue regeneration [31]. Therefore, ensuring an effective mechanotransduction response is essential, encompassing cellular inflammation and responses tied to vascular function. This entails the activation of macrophages adopting an anti-inflammatory phenotype (M2-like), characterized by the secretion of cytokines such as IL-10, as well as the release of growth factors such as PDGF, TGF-b, and VEGF. These signals play a pivotal role in promoting angiogenesis, facilitating the migration of keratinocytes, fibroblasts, and endothelial precursor cells and inducing the transition of fibroblasts into myofibroblasts. This orchestrated sequence of events contributes to the natural tissue remodeling process, culminating in the extracellular matrix being primarily composed of type I [30].

In the context of rehabilitation, the judicious choice of exercises, functional tasks, and mobility activities can play a key role in systematically and incrementally subjecting the knee structure to mechanical loading. This deliberate loading strategy serves to stimulate tissue healing and restorative processes. Through manipulation of the mechanical load parameters, healthcare professionals possess the means to influence tissue responses, facilitate motor skill acquisition, and rectify associated physical limitations. The provision of diverse loads that engender varying degrees of tensile, compressive, and shear deformation within the tissue via mechanotransduction and specificity offers a mechanism for fostering the requisite stress-induced adaptations. Such adaptations bolster tissue capacity and enhance its resilience against injury [31].

In this study, we propose tools such as strain SEL and cutaneous thermography as valuable instruments for the evaluation of the patellar tendon. These tools hold the potential for the oversight of rehabilitation training loads and the evaluation of the tendon’s responsiveness to mechanical stimuli. They, thereby, may enable the identification of excessive mechanical stresses and contribute to the facilitation of an optimal rehabilitation regimen. Additionally, it is important to consider the tissue’s load-bearing capacity, as evidenced by observations of adverse outcomes in low-load tissues such as ligaments in animal model studies [32].

The study by Dickson et al. showed altered tissue stiffness among individuals with knee osteoarthritis asymptomatic controls [24]. Sarikaya PZ et al. carried out a study on anterior shoulder instability after a labral tear where the structures of the shoulder were studied using elastography. It was observed that the rigidity of the structures was not considered a risk factor in anterior instability caused by trauma [33]. Wu CH et al. studied the stiffness of the shoulder ligaments at different degrees of shoulder mobility in people suffering from adhesive capsulitis [34]. In their study, they observed that the coracohumeral ligaments were stiffer in maximum external rotation than in the neutral position. They also observed that the coracohumeral ligament was thicker and stiffer in symptomatic than in asymptomatic shoulders [34].

This study presents certain strengths, namely (i) being the first to analyze the elastic changes in the PT (measured by SEL) along with its temperature and the relationship between both these variables as a result of a 448 kHz CRMR intervention in the patellar tendon in healthy physically active people, (ii) the group of participants had similar demographic characteristics and (iii) the SEL measurements were taken by an experienced professional, after a period of training, which guarantees the quality of the values obtained. In addition, the results provided may have an interesting application both in the field of injury prevention and in the clinical context, which opens new avenues of research. However, certain limitations must be recognized. The research was conducted on a small sample of healthy subjects which makes it difficult to extrapolate the results to other populations. Likewise, the analysis was performed on the patellar tendon, so extrapolations to other tendons should be made with caution. Ultrasound measurements, although widely accepted, used, and likely performed by novice evaluators presenting excellent reliability [35], are dependent on the operator, which must be taken into account when studying our results. Moreover, studies analyzing and reporting the reliability of SEL when assessing the patellar tendon should be carried out. Finally, as thermographic measurements are equally dependent on the operator, reliability should be further protocolized and studied; an appropriate sensor for close-range imaging with higher measurement accuracy, as well as utilizing specialized medical analysis software, would offer benefit, and even if a training protocol is carried out and the structure to be quantified is defined, the results must be interpreted with caution.

Immediate thermal changes might not necessarily translate into long-term clinical improvements. Thus, future longitudinal studies that analyze changes, in both healthy and diseased populations, after not only a CRMR intervention but also after a treatment program with short-, medium-, and long-term follow-ups, with and without exercise, as well as studying not only the patellar tendon but also other load tendons such as the Achilles tendon and plantar fascia, are necessary. In addition, other factors, especially intrinsic ones, should also be analyzed since they could alter the characteristics of the tendon. The potential impact of variations in participant positioning during the radiofrequency intervention on the distribution of the treatment’s thermal effects within the patellar tendon should also be further studied.

## 5. Conclusions

After one 448 kHz CRMR intervention there are changes in the temperature of the PT but not in its elastic properties. However, a relationship exists between the temperature of the PT and its elastic properties at the patellar insertion point after the CRMR intervention. More studies are needed to corroborate our findings in addition to longitudinal designs and standardized intervention programs to better understand the presented results.

## Figures and Tables

**Figure 1 diagnostics-13-02976-f001:**
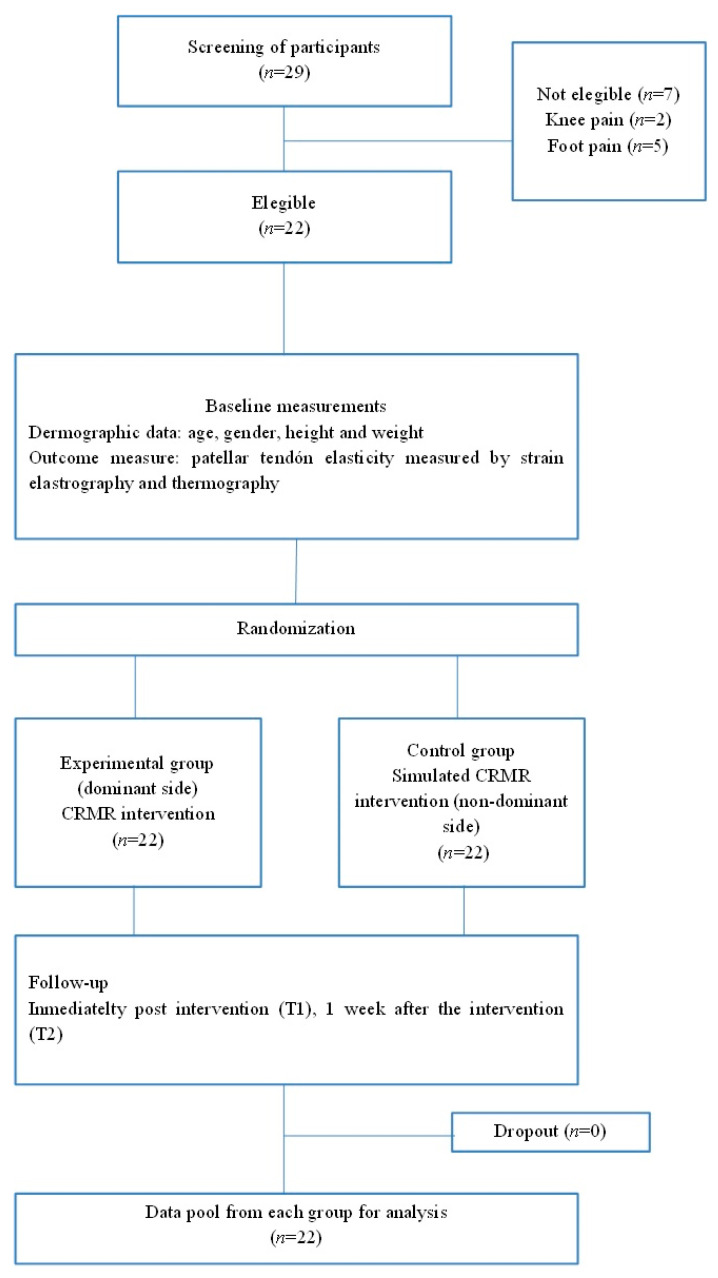
Flow diagram of participants.

**Figure 2 diagnostics-13-02976-f002:**
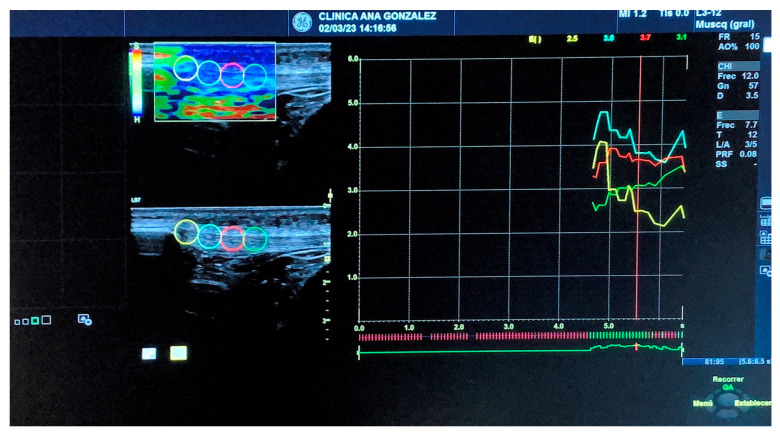
Patellar Tendon SEL measurements. Note. From left to right: Point 1: insertion of the PT to the kneecap (yellow); Point 2: body of the tendon in its med-proximal portion (blue); Point 3: body of the tendon in its mid portion (red); Point 4: body of the tendon in its mid-distal portion (green).

**Figure 3 diagnostics-13-02976-f003:**
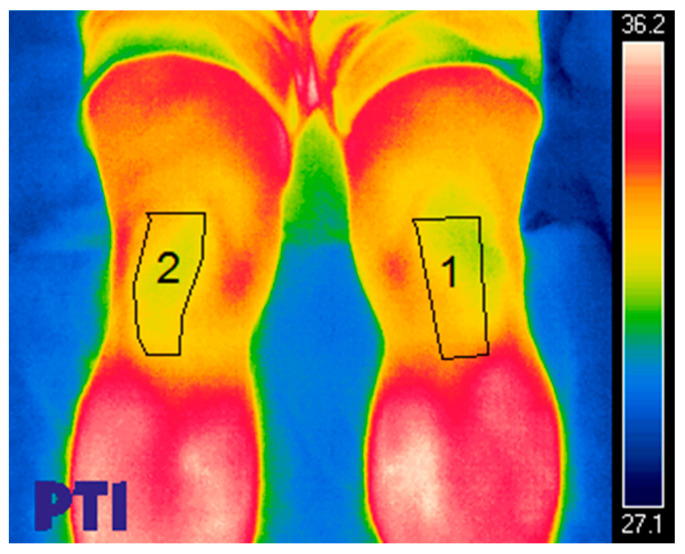
Thermographic assessment of the PT.

**Table 1 diagnostics-13-02976-t001:** Baseline demographic characteristics.

	Intervention Group (CRMR) (*n* = 22)	Control Group (Placebo CRMR) (*n* = 22)
Age (years), mean (SD)	33.6 (10.6)	33.6 (10.6)
Height (cm), mean (SD)	175 (8.51)	175 (8.51)
Weight (kg), mean (SD)	73 (10.4)	73 (10.4)
Patellar tendon elasticity point 1, mean (SD)	1.77 (1.09)	1.84 (0.695)
Patellar tendon elasticity point 2, mean (SD)	2.16 (1.14)	2.08 (0.695)
Patellar tendon elasticity point 3, mean (SD)	2.55 (1.10)	2.39 (0.907)
Patellar tendon elasticity point 4, mean (SD)	2.55 (0.995)	2.60 (1.23)
Patellar tendon thermography (SD)	32.1 (1.47)	32.2 (1.70)

CRMR: Capacitive resistive monopolar radiofrequency. SD: Standard deviation.

**Table 2 diagnostics-13-02976-t002:** Between-group differences in patellar tendon elasticity and temperature at baseline (T0); after the intervention (T1); and at one-week (T2) follow-up (95%CI).

	T0 (Baseline)	T1 (Immediately after the CRMR Intervention)	T1–T2 (Changes from Immediately after the CRMR Intervention until T2)	T2 (One-Week Follow-Up from T0)
Patellar tendon elasticity point 1 (mean)	0.07273 *p* = 1.000	0.16818 *p* = 0.982	0.01818 *p* = 1.000	−0.18636 *p* = 0.991
Patellar tendon elasticity point 2 (mean)	−0.0818 *p* = 1.000	0.2227 *p* = 0.954	−0.0364 *p* = 1.000	0.0909 *p* = 1.000
Patellar tendon elasticity point 3 (mean)	−0.1636 *p* = 0.994	0.0727 *p* = 1.000	−0.4500 *p* = 0.524	0.0364 *p* = 1.000
Patellar tendon elasticity point 4 (mean)	0.0500 *p* = 1.000	0.1909 *p* = 0.990	−0.4636 *p* = 0.622	0.0727 *p* = 1.000
Thermography (mean)	0.0227 *p* = 1.000	0.2273 *p* = 0.996	3.1591 *p* < 0.001	0.4091 *p* = 0.970

**Table 3 diagnostics-13-02976-t003:** Within-group differences in patellar tendon elasticity and temperature at baseline (T0); after the intervention (T1); and at one-week (T2) follow-up (95%CI).

	Intervention Group (CRMR)				Control Group (Placebo CRMR)			
	T0 (Baseline)	T1 (Immediately after the CRMR Intervention)	T1–T2 (Changes from Immediately after the CRMR Intervention until T2)	T2 (One-Week Follow-Up)	T0 (Baseline)	T1 (Immediately after the CRMR Intervention)	T1–T2 (Changes from Immediately after the CRMR Intervention until T2)	T2 (One-Week Follow-Up)
Patellar tendon elasticity point 1 (mean)	-	0.15000 *p* = 0.989	−0.33182 *p* = 0.838	−0.18182 *p* = 0.973	-	0.09545 *p* = 0.999	−0.35455 *p* = 0.797	−0.25909 *p* = 0.886
Patellar tendon elasticity point 2 (mean)	-	0.2591 *p* = 0.919	−0.0818 *p* = 0.999	0.1773 *p* = 0.966	-	0.3045 *p* = 0.852	−0.1318 *p* = 0.993	0.1727 *p* = 0.970
Patellar tendon elasticity point 3 (mean)	-	0.5227 *p* = 0.331	−0.1364 *p* = 0.989	0.3864 *p* = 0.521	-	0.2364 *p* = 0.937	−0.0364 *p* = 1.000	0.2000 *p* = 0.946
Patellar tendon elasticity point 4 (mean)	-	0.6545 *p* = 0.166	−0.2091 *p* = 0.923	0.4455 *p* = 0.583	-	0.1409 *p* = 0.995	−0.1182 *p* = 0.994	0.0227 *p* = 1.000
Thermography (mean)	-	−2.9318 *p* < 0.001	3.2727 *p* < 0.001	0.3409 *p* = 0.851	-	0.2045 *p* = 0.995	0.1818 *p* = 0.998	0.3864 *p* = 0.772

**Table 4 diagnostics-13-02976-t004:** Levels of association between patellar tendon strain elastography and thermography at baseline (T0); after the intervention (T1); and at one-week (T2) follow-up in both groups.

	Intervention Group (CRMR)			Control Group (Placebo CRMR)		
	T0 (Baseline)	T1 (Immediately after the CRMR Intervention)	T2 (One-Week Follow-Up)	T0 (Baseline)	T1 (Immediately after the CRMR Intervention)	T2 (One-Week Follow-Up)
Patellar tendon elasticity point 1	−0.242 *p* = 0.278	0.434 *p* = 0.044	0.031 *p* = 0.892	−0.047 *p* = 0.835	−0.346 *p* = 0.114	0.151 *p* = 0.503
Patellar tendon elasticity point 2	−0.352 *p* = 0.108	0.366 *p* = 0.094	0.005 *p* = 0.983	−0.009 *p* = 0.968	−0.405 *p* = 0.062	−0.189 *p* = 0.399
Patellar tendon elasticity point 3	−0.357 *p* = 0.103	0.053 *p* = 0.816	−0.247 *p* = 0.267	0.134 *p* = 0.553	−0.245 *p* = 0.272	−0.313 *p* = 0.156
Patellar tendon elasticity point 4	−0.329 *p* = 0.135	−0.081 *p* = 0.721	−0.365 *p* = 0.095	0.101 *p* = 0.654	−0.285 *p* = 0.198	−0.276 *p* = 0.213

## Data Availability

All data associated with this study are present in the paper. All requests for other materials will be reviewed by the corresponding author to verify whether the request is subject to any intellectual property or confidentiality obligations.

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
