# Peer review of "Patellar Tendon Elasticity and Temperature Following after a 448 Kilohertz Radiofrequency Intervention on Active Healthy Subjects: An Open Controlled Clinical Trial"

_diagnostics, 2023, doi:10.3390/diagnostics13182976_

Round 1
Reviewer 1 Report
Comments and Suggestions for Authors
The studies were well performed and the results are well presented. A few suggestions are given below for authors' considerations.
1. Line 25 and 227: "a low but significant..." may be better.
2. Line 52: Please provide a few sentences to describe the abnormal ultrasonic and elastographic findings of patients with patellar tendinopathy here.
3. Lin4 64-65: Please provide references here.
4. Line 77: What dose the acronym CRMR stand for here? I saw the full name on Line 186 but it is better to mention it when it appears the first time.
5. Line 230: What do you mean "they are" here? Other studies or the current study?
6. Line 241-242: The study of Dickson is not related to the current paragraphy. You may consider to move it to the next paragraphy. It will be even better if you can review other studies regarding the elastographic properties of different soft tissue, such as ligaments/tendons in shoulder joint.
Reviewer 2 Report
Comments and Suggestions for Authors
This is an open controlled clinical trial that investigated the effect of radiofrequency application at 4 locations at the proximal half of patellar tendons.
Line 49-52: Please use a reference pertaining to tendons and not to liver as these two tissues (i.e. liver vs tendon) are different in stiffness.
Line 56-57: Please provide the reference for the following statement "This technology is able to measure the relationship between body 56 temperature and muscle activation"
Line 79: Please reference original studies for spinal and knee OA.
Line 122: Please specify / reference which "tendon studies" are being referenced here.
Line 123: Please clarify what 1.1mm is referring to?
Line 132: Please clarify what perception this is referring to.
Line 143: Please specify years in numbers of use of ultrasound in clinical practice, with specific number of years of strain elastography use in practice.
Result section: Please stay consistent with significant digits/figures on all reported numbers.
Line 217: Please consider not re-stating the objectives of the study as this is already clear in previous sections.
Line 232 and 236: One may not know that the responses are necessary vascular. Please elaborate the reasons authors feel that the changes in elasticity/temperature are vascular in nature.
Discussion section: Please consider discussing your hypothesis on why there was a low and significant association at point 1 but not in other areas of patellar tendons. Also, please consider discussing if the resultant elasticity changes from this intervention is too small in magnitude for strain elastography to show changes.
Thank you for the opportunity to review this important work.
Author Response
Please see the attatchment.

Reviewer 3 Report
Comments and Suggestions for Authors
I´d like to express my sincere gratitude for the privilege of being selected as a reviewer for this article. I appreciate the opportunity to engage with this impactful research and offer my insights. These questions are aimed at stimulating deeper reflection and discussion regarding certain aspects of the study that could benefit from further clarification or exploration. They highlight areas where the authors could provide additional insights or considerations, enhancing the scientific rigor and practical applicability of the research:
1. Have you considered the potential impact of variations in participant positioning during the radiofrequency intervention on the distribution of the treatment's thermal effects within the patellar tendon? Given the intricate anatomical composition of the patellar tendon and the dynamic interaction of tissue structures during movement, it prompts the inquiry of whether the intervention's thermal effects can be consistently localized across different participant positions. This consideration becomes paramount as variations in tendon orientation could potentially influence the efficacy and safety of the treatment. The interplay between the radiofrequency's energy distribution and the tendon's complex biomechanical properties should be meticulously investigated to optimize the targeted therapeutic effects.
2. Given the known variations in tissue properties and healing responses across different tendons, do you think the findings regarding the patellar tendon's response to the CRMR intervention can be extrapolated to other tendons, like the Achilles tendon or plantar fascia? The investigation raises the broader question of whether the observed responses of the patellar tendon following a CRMR intervention can be confidently extended to other tendons with diverse anatomical locations, functions, and load-bearing characteristics. The unique biomechanical attributes, vascular supply, and tendon composition across distinct anatomical sites may contribute to nuanced responses. This underscores the importance of cautious extrapolation and motivates the exploration of tendon-specific adaptations to similar interventions.
3. Could you elaborate on the potential clinical implications of the observed relationship between patellar tendon elasticity and temperature? How might this information be translated into more targeted treatment strategies for individuals with patellar tendinopathy? The intriguing correlation between patellar tendon elasticity and temperature underscores the potential interplay between mechanical and thermal tissue properties. This intriguing connection invites speculation about whether a tailored therapeutic approach that modulates both mechanical and thermal factors could yield enhanced outcomes for individuals with patellar tendinopathy. Investigating whether interventions that synchronize mechanical loading and thermal modulation might amplify mechanotransduction and promote controlled tissue remodeling represents a promising avenue for optimizing clinical interventions.
4. Could you provide more details about the specific parameters of the 448 kHz CRMR intervention? For instance, how were the energy distribution and penetration depth determined, and how did these parameters align with the anatomical characteristics of the patellar tendon? The meticulous consideration of the specific parameters driving the 448 kHz CRMR intervention prompts a deeper inquiry into the rationale behind the chosen energy distribution and penetration depth. Aligning these parameters with the anatomical and biomechanical properties of the patellar tendon holds paramount importance. The investigation into the balance between energy delivery, tissue absorption, and depth of effect vis-à-vis the tendon's thickness and vascularization provides insights into the optimization of therapeutic efficacy while minimizing potential adverse effects.
5. Considering the potential for differences in tissue properties (e.g., collagen content, water content) across the patellar tendon's length, how did you ensure consistent and accurate measurements of strain elastography and thermography across the four different points along the tendon? Addressing these inherent heterogeneities could involve implementing rigorous quality assurance protocols to standardize imaging and measurement procedures. Moreover, delving into the potential impact of varying tissue characteristics on the reliability and interpretation of the obtained data provides valuable insights into the robustness of the findings.
6. Given that ultrasound measurements are operator-dependent, how did you address inter-operator variability in obtaining the strain elastography measurements? Were intra-observer reliability assessments conducted? Delving into the methodology adopted to assess and report the intra-observer reliability, along with its potential influence on the reported outcomes, illuminates the credibility and reproducibility of the findings.
7. The study seems to suggest that a single 448 kHz CRMR intervention leads to immediate thermal changes and some indications of vascular response. In the context of sports performance and recovery, could these changes have implications for athletes, particularly in terms of short-term recovery or enhancing tissue repair and adaptation?
8. Given the potential for bilateral adaptations following a unilateral intervention, could you discuss the potential impact of these contralateral thermal changes on overall athletic performance and injury risk? Delving into the mechanisms by which these bilateral adaptations may influence biomechanical symmetry, tissue responsiveness, and systemic physiological responses provides insights into potential implications for sports performance and injury mitigation. This consideration underscores the relevance of assessing how unilateral interventions might orchestrate systemic adaptations that impact an athlete's overall performance and susceptibility to injury.
9. While the study provides valuable insights into the acute effects of a 448 kHz CRMR intervention, have you considered discussing potential limitations or considerations for interpreting these results in a clinical context? For example, the immediate thermal changes might not necessarily translate into long-term clinical improvements.
10. In terms of the study's potential contributions to clinical practice, could you provide suggestions for how future research might build upon these findings? For instance, would it be valuable to investigate whether repeated CRMR interventions, combined with exercise, could lead to more sustained changes in patellar tendon properties?
11. In the study, an industrial-grade device, the FLIR T420bx Building IR camera, was used for human assessment, positioned 3 meters away from the subject, while medical ultrasound images were used closed to analyze 5 mm areas. Considering the sensor's field of view (FOV), what is the size of the area corresponding to each pixel in the thermal image? Inform, please. Moreover, with the patient in a seated position as shown in Figure 2, where inclined images are obtained, and considering the device's low accurate temperature measurements of +/- 2°C, how did the authors address the reliability of the results obtained through thermography? The software employed does not seem suitable for this study, as it appears to be designed for distant assessments without concern for the details of 5 mm changes obtained through ultrasound, which involves direct patient contact. An assessment with such a distant sensor does not seem consistent with the medical radiology logical approach. It appears that the authors could have benefitted from experienced specialized guidance in selecting a more appropriate sensor for close-range imaging with higher measurement accuracy, as well as utilizing specialized medical analysis software for this study. Could you provide insights into your considerations for these aspects and explain them in the text?
Comments on the Quality of English LanguageHere are some improvements and corrections for errors in the English language that could be made in the article:
- Title:
- Original: "Elastographic and Thermographic Changes of the Patellar Tendon After 448 kHz CRMR Intervention: A Controlled Clinical Trial"
- Suggestion: "Elastographic and Thermographic Changes in the Patellar Tendon Following a 448 kHz CRMR Intervention: A Controlled Clinical Trial"
- Introduction:
- Original: "Thanks to its morphology and adaptive properties, it has the ability to respond to intrinsic and extrinsic stimuli."
- Suggestion: "Due to its morphology and adaptive properties, the tendon exhibits the capacity to respond to both intrinsic and extrinsic stimuli."
- Introduction:
- Original: "Patellar tendinopathy is usually accompanied by pain in the anterior aspect of the knee, at the lower pole of the kneecap."
- Suggestion: "Patellar tendinopathy is typically characterized by anterior knee pain, localized at the lower pole of the patella."
- Introduction:
- Original: "vascular, mechanical, or impingement-related causes"
- Suggestion: "vascular, mechanical, or causes related to impingement"
- Introduction:
- Original: "asymmetry between different body areas"
- Suggestion: "asymmetry among various body regions"
- Methods:
- Original: "baseline demographic and clinical data were collected."
- Suggestion: "baseline demographic and clinical information was gathered."
- Methods:
- Original: "Participants who presented: (i) any type of pain or inflammatory process"
- Suggestion: "Participants presenting: (i) any form of pain or inflammatory process"
- Methods:
- Original: "informed written and verbal consent"
- Suggestion: "informed written and verbal consent was obtained"
- Results:
- Original: "significant differences in terms of age, gender, height, weight and PT elasticity"
- Suggestion: "significant differences in age, gender, height, weight, and PT elasticity"
- Results:
- Original: "statistically significant differences in the patellar tendon temperature"
- Suggestion: "statistically significant differences in patellar tendon temperature were observed"
- Results:
- Original: "thermal changes just after the 448 kHz CRMR intervention."
- Suggestion: "thermal changes immediately following the 448 kHz CRMR intervention."
- Discussion:
- Original: "there is an immediate vascular response in the knee region after the application of 448kHz CRMR"
- Suggestion: "an immediate vascular response occurs in the knee region following the application of 448 kHz CRMR"
- Discussion:
- Original: "tendon stiffness between individuals with knee osteoarthritis and asymptomatic controls"
- Suggestion: "tendon stiffness differences among individuals with knee osteoarthritis and asymptomatic controls"
It's advisable to have a professional editor or a native English speaker review the entire article for further enhancements.
Author Response
Please see the attachemnt.

Round 2
Reviewer 3 Report
Comments and Suggestions for Authors
I deeply appreciate the meticulous effort you and your co-authors have invested in addressing my comments and enhancing your manuscript. Your comprehensive and thoughtful responses reflect a commendable dedication to refining the paper, as evidenced by the careful consideration of each point, substantial improvements made, and a proactive approach to implementing suggestions that enhance scientific rigor and clarity. Your commitment to engaging with my feedback and incorporating constructive changes is truly praiseworthy. Thanks.